# Quicklime and Calcium Sulfoaluminate Cement Used as Mineral Accelerators to Improve the Properties of Cemented Paste Backfill with a High Volume of Fly Ash

**DOI:** 10.3390/ma13184018

**Published:** 2020-09-10

**Authors:** Hangxing Ding, Shiyu Zhang

**Affiliations:** 1Key Laboratory of Ministry of Education on Safe Mining of Deep Metal Mines, Northeastern University, Shenyang 110819, China; dinghangxing@mail.neu.edu.cn; 2Science and Technology Innovation Center of Smart Water and Resource Environment, Northeastern University, Shenyang 110819, China

**Keywords:** calcium sulfoaluminate cement, quicklime, fly ash, hydration evolution, CO_2_ emission

## Abstract

In order to reduce the CO_2_ emission and cost of binders used in cemented paste backfill (CPB) technology, new blended binders with a large amount of fly ash (FA) were fabricated. Different doses of quicklime and calcium sulfoaluminate cement (CṠA) were used as mineral accelerators to improve the early workability of CPB. The effects of CṠA and quicklime on flowability, compressive strength, pore structure, hydration heat, and hydration evolution were investigated experimentally. The results showed that the addition of quicklime and CṠA reduced the spread diameter of the fresh backfill and improved the mechanical performance of the hardened CPB. With increasing quicklime and CṠA, the cumulative hydration heat of the blended binder distinctly increased in the first 6 h. CṠA improved the initial hydration by increasing the reactivity, and quicklime increased the hydration rate by activating FA. The blended binder (15% quicklime + 10% CṠA) with the lowest CO_2_ emission and cost had potential application in filling technology.

## 1. Introduction

Cemented paste backfill (CPB) technology is widely accepted in the mining industry, addressing the issue of instability in underground stopes and mine tailings [1,2,3,4,5]. This technology correspondingly solves many issues (low ore recovery, surface subsidence, acid mine drainage, groundwater pollution, tailing dam failure, etc.) and brings economic and environmental benefits [6,7,8,9,10,11]. CPB is an engineered mixture composed of thickened and filtered tailings, water, and hydraulic binder, with a total solid concentration (tailings + binder) of 70–85% [7,12,13,14,15,16,17,18,19]. The function of the binder is to maintain a uniform suspension of tailings during pipe transportation, allowing the hardened CPB to achieve the required strength [4,7]. The ratio is in the range of 2–7% (sometimes up to 10%), and the binder accounts for about 75% of the total filling cost [3,20,21].

In recent years, discussions on saving filling costs have become mainstream. Much attention has been directed to the potential use of aluminosilicate source of materials (i.e., fly ash (FA), blast-furnace slag, metakaolin, and silica fume) as a partial replacement for cement in backfill. FA, a by-product of power plants, is commonly used as supplementary cementitious material for fabricating cement-based materials. The addition of FA reduces the rheological properties and improves the flowability of the fresh backfill due to the lubrication effect of spherical morphology [22]. Besides, FA, rich in aluminum and silicon, can be involved in cement hydration and form hydrate products (e.g., calcium silicate hydrate, calcium aluminate hydrate, and ettringite) that affect strength development [23,24,25,26]. Deschner et al. [27] documented that with an early hydration time of up to 2 days, cement hydration is mainly affected by the so-called “filler effect” of FA. At the same time, the FA reaction is detected by the consumption of portlandite after a hydration time of 2 days. Schöler et al. [28] showed that low-calcium FA slightly reduced the formation of portlandite and calcium silicate hydrate but increased the formation of monocarbonaluminate/hemicarbonaluminate. Addition of less than 30% FA has a limited effect on strength development at later ages. However, the mechanical performance of the backfill with high volume (>30%) of FA is significantly reduced due to low activity, especially at an early age.

To overcome this issue, several methods have been used to improve the mechanical properties of CPB in the early hydration time. Strong alkaline activators, which are generally composed of sodium/potassium hydroxide (NaOH/KOH) and sodium/potassium water glass (nSiO_2_·Na_2_O) in fixed proportions, are an effective option. The high pH of the alkaline solution can break down covalent Si–O–Si and Al–O–Si bonds, providing more dissolved silica and aluminum to for hydration reaction [29,30]. However, the production of NaOH/KOH and nSiO_2_·Na_2_O consumes a lot of energy and can release an immense amount of CO_2_, so it does not lead to environmental protection. The use of quicklime, a weakly alkaline mineral promotor, is a relatively simple and economical alternative. Shi [31] first investigated the effect of quicklime on natural pozzolan/cement systems in 2001, with promising results regarding the early-stage promoting effect of quicklime. Antiohos et al. [32,33] experimentally explored the influence of industrially produced quicklime on the pozzolanic reaction rates of various fly ash/cement systems. An addition of 5% quicklime was found effective only during the initial hydration stage, and almost no accelerating effect was detected with an increased dose. However, these studies have primarily focused on a small addition of FA. Quicklime activation is not enough for strength development of a cementitious system, with FA content exceeding 30%.

Calcium sulfoaluminate (CṠA) cement has been widely used in concrete repair engineering due to its high initial compressive strength, microexpansion, low-temperature adaptability, and poor permeability [34,35]. Ye’elimite is the main mineral component of CṠA. After hydration, it forms monosulfoaluminate (AFm) or ettringite (AFt) based on the concentration of sulfate [34,35,36]. The rapid hydration of ye’elimite shortens the setting time and increases the initial strength rapidly. All these advantages support the utilization of CṠA and large amounts of fly ash as supplementary cementitious materials in CPB. However, previous studies failed to evaluate the synergistic effect of quicklime and CṠA on the properties of CPB with high amounts of FA.

Therefore, in this paper, CṠA and quicklime were used as mineral promotors to increase the hydration rate of cementitious systems. Slag cement (SC), which consists of cement clinker, granulated blast-furnace slag, and gypsum, was selected as a basic cementitious material because it is lower in cost and emits less CO_2_ than Ordinary Portland cement (OPC). Fluidity and unconfined compressive strength (UCS) tests were conducted to evaluate the workability of CPB. The hydration evolution of solid phases was determined using several laboratory techniques to suggest a mechanism for experimental findings.

## 2. Materials and Methods

### 2.1. Raw Materials

The basic cementitious material used in this study was PSA 3.25 SC (Shandong Juzhou Cement Co., Ltd., Rizhao, China). The particle size distribution and chemical composition of SC were measured by Malvern Mastersizer 2000 (Malvern Instruments Co. Ltd., Malvern, UK) and X-Ray Fluorescence Spectrometer, respectively. Figure 1a shows that the SC of D_20_, D_50_, and D_80_ are 6.72, 18.7, and 43.2 μm, respectively. D_20_, D_50_, and D_80_ represent the critical diameters when the cumulative volume of solid particles reaches 20%, 50%, and 80%, respectively. The summarized results of chemical composition in Table 1 show that SC contains 55.24% calcium oxide (CaO), 26.15% silicon dioxide (SiO_2_), and 8.83% aluminum oxide (Al_2_O_3_). The mineral phases of SC were identified by XRD and are shown in Figure 2. As shown in the figure, SC is rich in crystalline calcite, quartz, mullite, tricalcium silicate, and dicalcium silicate.

Despite its high cost, CṠA (Shandong Juzhou Cement Co., Ltd., Rizhao, China) was used as supplementary cementitious material (limited addition) due to its superior gelling property. The physical and chemical properties were characterized and shown in Figure 1a and Table 1, respectively. The D_20_, D_50_, and D_80_ of CṠA are 8.97, 18.36, and 31.1 μm, respectively. It can be concluded that CṠA has a relatively narrow particle size distribution compared to SC. The main chemical composition of CṠA is 53.91% CaO, 13.6% SiO_2_, 15.13% Al_2_O_3_, and 10.35% SO_2_. The identified mineral phases of CṠA shown in Figure 2 are ye’elimite, anhydrite, calcite, tricalcium silicate, and dicalcium silicate. In addition, quicklime (Liaoning Bangka Calcium Industry Co., Ltd., Benxi, China) was also adopted here as a mineral activator.

FA was used as a supplementary cementitious material to replace 40% SC. FA has a coarser particle size with the D_20_, D_50_, and D_80_ of 12.06, 34.73, and 82.35μm, respectively. It contains 55.51% SiO_2_, 30.82% Al_2_O_3_, and only 4.08% CaO. XRD analysis results indicate that FA is composed of crystalline quartz, mullite, and magnetite. In order to eliminate the influence of tailing minerals on the properties of CPB, silica tailings (ST) with 96.46% SiO_2_ was used as aggregate. ST has the D_20_, D_50_, and D_80_ of 28.45, 58.09, and 93.21 μm, respectively. The morphology of SC, CṠA, quicklime, and FA was detected using SEM. As shown in Figure 3, FA has a spherical shape, and SC, CṠA, and quicklime have irregular shapes.

### 2.2. Sample Preparation

Six groups of pure blended binders were prepared according to the detailed mixing ratios shown in Table 2. The first sample series was prepared to explore the influence of CṠA on hydration reactions. The mass ratios of SC:CṠA:FA:quicklime were 10:0:8:2, 9:1:8:2, and 7:3:8:2, and the sample numbers were defined as SCA0, SCA5, and SCA15, respectively. A second sample series was prepared to confirm the effect of quicklime on the hydration reaction. The mass ratios of SC:CṠA:FA:quicklime were 9:2:8:1, 8:2:8:2, and 7:2:8:3, and the sample numbers were defined as SCC5, SCC10, and SCC15. A control sample was also prepared and named SC100 for comparison. The water/binder ratio was 0.43. Accordingly, seven groups of CPB samples were prepared using blended binders. The binder/ST ratio was 0.2, and the solid concentration was 70% by mass. After mixing for 5 min, a portion of the fresh backfill was subjected to the fluidity test, and the remaining was poured into plastic cylinder molds (5 cm in diameter × 10 cm in height). It should be noted that both of the upper and lower ends of the molds are sealed. All samples were cured at room temperature (20 ± 2 °C) with a relative humidity of 95 ± 2%.

### 2.3. Testing Methods

#### 2.3.1. Spread Diameter

The slump test is convenient and is often used to assess the workability of fresh concrete. Recently, this method has been increasingly adopted to measure the fluidity of fresh backfills. According to the literature [11], the spread diameter (SD) obtained from the slump test is determined by the yield shear stress, an important rheological parameter of fluids. Hence, in this study, a mini-cone (5 cm in top diameter, 10 cm in bottom diameter, and 15 cm in height) was selected to measure the diameter of the spread.

#### 2.3.2. Unconfined Compressive Strength

After reaching the pre-determined hydration time (3, 7, and 28 days), CPB samples were subjected to the UCS test based on ASTM C39 standard test procedure [37]. A computer-controlled loading machine (Humboldt HM-5030(Raleigh, NC, USA)) was used herein. The load capacity was 50 kN, and the load-deformation rate was 1 mm/min. Both ends of the cylindrical specimen were kept as flat as possible to reduce edge effects. During the compressive test, the peak stress corresponds to the mechanical strength of the sample. If the measurements of the two samples differ by more than 15%, a third sample needs to be measured to verify its accuracy. Hence, at least two samples were measured for each recipe, and only the mean value was considered the UCS for CPB samples.

#### 2.3.3. Pore Structure

The pore structure of the samples was analyzed by the MIP method using Micromeritics’ AutoPore IV 9500 (Atlanta, GA, USA). A piece of sample less than 15 × 15 × 15 mm in volume was taken from the crushed specimen whose location must be far enough from the shear plane [19]. Samples were immersed in isopropanol to stop the hydration for 12 h, and then dried in vacuum drying oven at 45 °C for 24 h. The testing range for pore diameter was 3–1000 μm.

#### 2.3.4. Hydration Heat

Approximately 6 g of binder paste with a water/binder ratio of 0.43 was loaded into a glass vial and mixed evenly using a slow stirring mixer. In this study, isothermal calorimetry was conducted using a TAM Air 8-channel microcalorimeter (New Castle, DE, USA) to confirm the heat flow of the blended binder. The ambient temperature was 20 ± 2 °C, and 72 h of experimental data was recorded.

#### 2.3.5. Hydration Evolution

As discussed in the literature [25,26,38,39], the hydration process of pure binder pastes cured for 3, 7, and 28 days, respectively, was stopped by isopropyl alcohol. Samples were pulverized after drying in a vacuum drying oven to a constant weight at 45 °C. Several techniques were employed to characterize the hydration evolution:

(1) XRD patterns of binder pastes were recorded using a XRD 7000 diffractometer (Tokyo, Japan). The test interval angle was 5–50°, and the scan step size was 5°/min.

(2) Phase composition analysis of binder pastes was performed using thermogravimetry analysis STA409PC (Selb, Germany) coupled with differential thermogravimetry (DTG). About a 30 mg sample was loaded into an alumina crucible and heated to 1000 °C in a nitrogen atmosphere (15 °C/min) [25,26]. The amount of hydrate water (H) and portlandite (CH) can be expressed as:(1)$$H\;=\;\frac{M_{50}-M_{550}}{M_{550}}\cdot 100\%$$
(2)$$\mathrm{CH}\;=\;\frac{M_{400}-M_{550}}{M_{550}}\cdot \frac{74}{18}\cdot 100\%$$
where *M*_50_, *M*_400_, and *M*_550_ correspond to the residual mass of samples at temperatures of 50 °C, 400 °C, and 550 °C, respectively.

(3) A SEM (Hitachi S-3400N (Tokyo, Japan)) was used to observe the morphology of samples at an accelerating voltage of 15 keV. Energy-dispersive X-ray (EDX) spectroscopy was used to analyze the elemental composition. More details can be found in reference [40].

## 3. Results

### 3.1. Spread Diameter

Figure 4 shows the effect of the addition of quicklime and CṠA on the flowability of fresh CPB. The addition of CṠA slightly increased SD, but when SC was replaced with quicklime, the fluidity decreased to some extent at the initial setting time of 5 min. For instance, SD increased from 29.47 cm to 30.86 cm with an increment of 1.39 cm for the samples with the addition of CṠA increasing from 0 to 15%, but decreased from 32.41 cm to 26.73 cm with a reduction of 5.68 cm for the samples with addition of quicklime increasing from 5% to 15%.The SD gain of fresh sample is attributed to the large amount of ultrafine particles with a diameter of less than 20 μm in CṠA. Partial replacement of the SC increased the packing density of the fresh sample. According to the literature [41], the amount of free water that imparts fluidity to a fresh slurry is positively related to its packing density. In this case, the addition of CṠA increased the amount of free water for flowability. Additionally, the decrease in SD with increasing doses of quicklime is mainly associated with water-consuming chemical reactions and irregular shapes, as shown in Figure 3. Regardless of the effects of CṠA, quicklime, or both, the fluidity performance of fresh samples with 40% FA is better than with pure SC. This is due to the lubrication effect of spherical FA (Figure 3).

The SD of all fresh samples clearly decreased with increased curing time. This is associated with the initial hydration reaction. Tricalcium silicate, tricalcium aluminate, and calcium sulfoaluminate quickly dissolved in water and hydrated to form AFt/AFm and gel products [36]. The reduction in SD is based on the doses of CṠA and quicklime added. In the SC100 sample, the SD experienced a drop of 5.23 cm with the hydration time increasing from 5 to 60 min. With the addition of 15% CṠA and 10% quicklime (SCA15), SD reduced by 14.42 cm. Ye’elimite, the most reactive component of CṠA, reacts with water to produce cementitious materials in a short time, promoting the solidification of fresh CPB significantly. Increasing the amount of quicklime from 5% to 15% resulted in similar reductions in SD for SCC5, SCC10, and SCC15 with increased setting time, higher than the SC100 samples. This is because the dissolution of the quicklime that is associated with releasing heat can partially increase the hydration rates of silicates, aluminates, and sulfoaluminates.

### 3.2. Strength Development

Figure 5 shows the strength development of CPB hardened with different doses of CṠA and quicklime. It was found that when the amount of FA (40%) that replaces SC was large, the mechanical performance noticeably decreased irrespective of the curing time. In contrast, the addition of CṠA and quicklime had a positive effect on the mechanical properties. As shown in Figure 5a, compared with SCA0, the addition of 5% CṠA (SCA5) caused a 0.21 MPa increase over a 3-day curing time, and a 15% SC replacement (SCA15) caused a 0.39 MPa increase. This inconsistent improvement in strength (nearly twice) relative to the amount of CṠA added (three times) is potentially attributed to insufficient drainage due to more ultrafine particles (Figure 1a) [3]. As shown in Figure 5b, the UCS value of samples with 10% SC replacement (SCC10) increased from 0.53 MPa to 0.66 MPa with an increase of 0.13 MPa in the hydration time of 3 days, while the addition (SCC15) of 15% quicklime caused an improvement of 0.1 MPa. When quicklime reacts with water, it releases a large amount of hydroxyl ions, leading to an alkaline environment and activating FA [32]. Amorphous silicon and aluminum liberated by activated FA particles consume portlandite to anticipate in hydration reaction, producing ettringite and C–S–H/C–A–S–H. The addition of a 10% quicklime can accelerate the pozzolanic reaction of FA and compensate for the loss of hydration products due to the replacement of 10% SC. When the replacement of SC with quicklime is increased to 15%, the limited solubility of calcium hydroxide in water restricts the acceleration effect of FA on the pozzolanic reactions.

It was also found that the improvement in curing time from 7 to 28 days compared to the increase in UCS at seven days of hydration time was approximately similar regardless of binder types. In SC100 samples, SC consists of Portland cement clinker, 20–50% granulated blast furnace slag, and an appropriate amount of gypsum. Coupled with the filler effect of granulated blast-furnace slag particles, ettringite and gel products (e.g., C–S–H), which result from the hydration of aluminates and silicates, strengthen the hardening process at the curing time of 7 days. Although slag is highly active and can quickly participate in the hydration reaction, it does not perform well in the subsequent hydration process compared to the hydration rate of cement clinker. This results in a slow increase in strength in the next 7 to 28 days of curing time. By adding CṠA and quicklime in the blended binders, ettringite develops as the main hydration product and fills the voids that exist in solid skeletons. With extra sulfate ions, quicklime can be hydrated to form dihydrate gypsum, which also leads to the hardening of CPB [42]. However, when sulfate ions are consumed, monosulfoaluminate is formed instead of ettringite [43,44].As a result of low crystallinity, the filler effect of AFm on CPB is low [25]. Furthermore, FA activated by quicklime hydration participates in the hydration reaction and decreases pH with increasing curing time. This is not desirable for subsequent pozzolanic reactions [42,45,46,47]. Hence, inadequate hydration of blended binders causes a slow increase in mechanical performance within the hydration time of 7–28 days.

### 3.3. Pore Structure

Figure 6 and Figure 7 show variations in the incremental pore volume curves of CPB at various doses of CṠA and quicklime over a 28-day curing time. As shown in the figures, the pore size distribution of the SCA0-15 and SCC5-15 samples is relatively scattered compared to the SC samples. This is mainly associated with the filler effect of FA and leads to the refinement of the pore structure [48]. However, large amounts of FA cannot react completely, resulting in lower content of hydration products (e.g., AFt, AFm, and C–S–H/C–A–S–H) and higher pore volumes. This can be used to explain the weak mechanical performance of CPB with 40% FA, as shown in Figure 5. In addition, it was clearly observed that the increased content of CṠA and quicklime contributed to the improvement of the pore structure and decreased its total volume. This is because CṠA hydrates rapidly, and hydrated quicklime activates the pozzolanic reaction of FA, as explained in Section 3.2. Additional hydration products can be formed to fill the voids.

Based on the references [49,50,51,52], pores with diameter smaller than 500 nm are micropores and mespories, and pores with diameter larger than 1000 nm are large capillary pores. Hence, pores are classified into three types according to the pore size in this study: (1) type I: < 500 nm; (2) type II: 500–1000 nm; (3) type III: > 1000 nm. The volumes of the three categories of pores are also listed in Figure 6 and Figure 7. As shown in the figures, SC100 samples had the largest volume of type I pores and the smallest volume of type III pores. Pores with diameters greater than 1000 nm are called “harmful pores” that significantly influence strength development [53,54,55]. This verifies the result that SC100 backfill has the best mechanical performance. As the amount of CṠA and quicklime increased, the volume of the type III pores decreased dramatically, which was associated with the hydration reaction rate. The addition of CṠA and quicklime accelerates the hydration and thus improves the mechanical properties of hardened backfill. In order to investigate the relationship between pore volume and mechanical strength, a linear regression between strength and type I pore volume, type II pore volume, type III pore volume, and total pore volume was calculated (Figure 8). It could be found easily that UCS of samples had a positive relationship with the volume of type I pores, while a negative relationship existed between the strength and type III pore volume. This can be explained by the influence of “harmful pores”. Their corresponding correlation coefficients are 0.521, 0.235, 0.654, and 0.887, indicating that the total pore volume (porosity) is responsible for the mechanical performance of hardened CPB.

### 3.4. Hydration Heat

Figure 9 shows the normalized heat flow of blended binder pastes, and SC100 (control) is included for comparison. It can be clearly seen that the first peak of the heat flow of binder pastes appeared in the first 30 min (Figure 9a). This is mainly attributed to the dissolution of soluble particles, including cement clinker, ye’elimite, and quicklime [27]. The SC100 paste released about 0.014 W/g of heat mostly due to the wetting of tricalcium aluminate (C_3_A) and the initial precipitation of ettringite. When SC was replaced with 40% FA, the exothermic peak value showed a significant decrease. Coupled with the reaction of quicklime and water, increasing the dose of CṠA from 0% to 15% increased the heat peak from 0.0095 to 0.013 W/g. This increase results from faster dissolution and hydration of ye’elimite. Increasing the dose of quicklime from 5% to 15% increased the first exothermic peak from 0.01 to 0.022 W/g with the accelerating effect of CṠA. This improvement is due to the more vigorous reaction of quicklime forming portlandite. Then, as the concentration of calcium, sulfate, and hydroxyl ions increased, a decrease in heat flow occurred [42]. The initial alkaline environment can be strengthened by the increased amount of portlandite generated, promoting the activity of FA [56]. The dissolution of amorphous silicon and aluminum from active FA contributes to the heat flow. Hence, the SCC15 sample exhibited the highest rate of heat release in the first 1 h.

An exothermic acceleration phase appeared after 4h, which is associated with the dissolution of tricalcium silicate (C_3_S) and anhydrite, synchronous with the precipitation of portlandite and C–S–H phases [57]. With the increased formation rate of ettringite, the concentration of sulfate ions decreased, leading to the release of sulfate ions originally attached to tricalcium aluminate. This leads to a further dissolution and hydration reaction of tricalcium aluminate [57]. None of the heat flow curves for the blended binders containing additional FA showed sharp peaks, but wider shoulders. This different behavior is associated with the retarding effect of FA reported in the literature [58,59]. Cumulative heat release from the first 6 h of hydration was higher for all blended binders than the SC100 paste, excluding the SCA0 sample. This indicates that the hydration rates of CṠA and quicklime cover the insufficient hydration due to the replacement of cement with FA. The cumulative heat release of the SC100 sample showed the most noticeable increase with increasing hydration time from 6h to 12 h. This is associated with a higher content of cement clinker (e.g., C_3_A and C_3_S). With a curing time of 72 h, the SC100 sample released the highest amount of heat, and the SCC15 paste ranked second. The additional heat of SCC15 compared to other blended binders is derived from a large amount of activated FA particles involved in the hydration reaction.

### 3.5. Hydration Evolution

#### 3.5.1. XRD Patterns

The XRD patterns of blended binders and control samples were measured at hydration times of 3, 7, and 28 days (Figure 10). The quartz phase of all blended binders was higher than the SC100 sample. This is mainly related to replacing SC with large amounts of FA, which is rich in crystalline quartz (Figure 2). As the amount of CṠA increased from 0% to 15%, the intensity of the ettringite peak became more pronounced. Ye’elimite (C_4_A_3_Ṡ) reacts with water in the presence of gypsum to form ettringite. The H_c_/M_c_ peak was detected, and the intensity of this peak strengthened with increasing curing time. C_3_A and C_4_A_3_Ṡ react with calcium carbonate to form M_c_ [43,44]. Due to the consumption of calcium carbonate, insufficient carbonate leads to the formation of H_c_ instead of M_c_. In addition, the value of ettringite peak increased with increasing curing time from 3 to 28 days. The formed H_c_/M_c_ inhibits the conversion of AFt to M_s_. It was impossible to detect the M_s_ phase by XRD due to poor crystallinity. The peaks of C–S–H and calcium carbonate in the SC100 sample were higher than all blended binders, irrespective of curing time. This implies that SC100 has a faster hydration rate than blended binders. This is consistent with the strength development shown in Figure 5. When the dosage of quicklime increased from 5% to 15%, the portlandite phase was intensified. In addition, dicalcium silicate (C_2_S) was also detected in all samples. This is because less reactive C_2_S hydrates in the late age.

#### 3.5.2. TG Analysis

To characterize the transformation of hydration products formed, TG analysis was performed on all binder pastes in curing times of 3, 7, and 28 days. DTG curves were obtained by the first derivative of the TG data, as shown in Figure 11. Weight loss dehydration of ettringite almost occurred at 75–120 °C [34]. Thus, it is clearly seen that all blended binders, except the SCA0 (0% addition of CṠA) and SCA5 (5% addition of CṠA) samples, showed higher peak values than SC100 samples after hydration for three days. This indicates that more AFt was generated early in hydration. The formation of AFt probably results from the fast hydration of ye’elimite in the presence of gypsum. This contributes to the filler effect and improves the strength of the hardened backfill. Besides, the H_c_/M_c_ peak value for the SC100 sample was found to be higher than blended binders, regardless of curing time. This may be related to the high content of C_3_A in neat SC.

According to the literatures [40,60,61], the decomposition temperatures of hydration products (AFt, AFm, and C–(A)–S–H) and portlandite are 50–550 °C and 400–550 °C, respectively, while the decomposition of carbonate mainly appears between 550–1000 °C. The amount of hydrate water and portlandite obtained is shown in Figure 12. As can be seen in the Figure 12a, the SC100 sample contained the most chemically bound water (hydrate water) during the hydration process and, therefore, the highest hydration products, regardless of the curing time. As the dose of CṠA increased from 0 to 15%, the amount of hydrate water increased dramatically. This is because ye’elimite is highly reactive. When the amount of quicklime added was increased from 5% to 15%, changes in hydrate water showed a similar upward trend. This increase in hydration products is a result of the addition of more quicklime to create a highly alkaline environment. The high concentration of hydroxide ions promotes the breakdown of covalent Si–O–Si and Si–O–Al bonds in FA. At the same time, negatively charged tetrahedral anions (SiO_4_^4−^ and AlO_4_^5−^) absorb positively charged ions (e.g., Ca^2+^) to drive polycondensation under the action of hydration [56]. With sufficient portlandite and sulfate, ettringite and C–S–H/C–A–S–H gelling products are formed. In addition, the increase in hydration products is also partially due in part to the hydration of quicklime, which produces portlandite.

The SC100 sample presented an increase in portlandite formed with longer curing time, indicating a higher degree of hydration. For blended binders, the content of portlandite tended to increase and then decreased from the hydration time of 3 to 28 days. With a 7-day curing time, a small fraction of FA is activated and participates in the hydration reaction. The newly formed portlandite from cement hydration exceeds the amount consumed in the FA reaction, while after a 7-day curing time, a large amount of FA participates in the hydration reaction and consumes a considerable amount of calcium hydroxide. The SCC15 sample had more portlandite than SCC10 paste. Combined with the TG results and the strength development results, it can be inferred that the addition of 15% of quicklime reached the limit.

#### 3.5.3. SEM Observation

SEM observation of the microstructure of blended binders was performed, and the results are shown in Figure 13. A control sample (SC100) was also included for comparison. As shown in Figure 13, the SC100 sample has a relatively coarser microstructure than blended binders, which reveals a higher degree of hydration. Increasing the amount of CṠA from 0 to 15% caused the morphology of samples to be rough, which is related to the acceleration effect of CṠA. Increasing the amount of quicklime from 5 to 10% caused a clear change in the microstructure of SCC10. As the dose of quicklime continued to increase to 15%, the morphology did not continue to coarsen, and there was a large amount of calcium hydroxide. This is consistent with TG experimental results shown in Figure 12.

EDX analysis was also conducted to evaluate the C–S–H composition of binder pastes, and the results are shown in Figure 14. According to the literatures [60,62,63], the Al/Ca atomic ratio of portlandite, AFt, and AFm are 0, 0.33, and 0.5, respectively. To avoid the influence of AFt and AFm, a straight line was drawn through the points with the lowest Al/Ca ratio to obtain the Al/Si atomic ratio. As seen in Figure 14, the SC100 sample had an Al/Si atomic ratio of about 0.14, and a Si/Ca atomic ratio of about 0.3–0.4. When incorporated into FA, CṠA, and quicklime (blended binder), the data points were dispersed. This is mainly attributed to the variable composition of active FA involved in the hydration reaction. In addition, if the Si/Ca ratio is extremely high, it may be caused by the measurement error due to wrong dot position. There is a trend toward an increase in the overall Si/Ca ratio for FA-cement cementitious system, which is consistent with the results of previous studies [64,65]. Blended binders had higher Al/Si atomic ratios (0.27 and 0.43) than the control sample. This means that more FA is activated and hydrated to form C–S–A–H. C–S–H absorbs dissolved Si to form longer silicate chain lengths, allowing the bridging tetrahedra of silicate chains to absorb more aluminum [66,67].

### 3.6. Discussion

This study developed novel blended binders composed of SC, FA, quicklime, and calcium sulfoaluminate cement. SC provides major gel products as basic cementitious material. FA has the potential to reduce CO_2_ emissions and costs as a source of aluminosilicate material. The filler effect of FA provides additional nucleation sites for cement hydration, increasing the packing density of the cementitious system, and refining the pore structure [40,68,69]. Under the influence of the alkaline environment, active FA particles participate in hydration to form C–S–H/C–A–S–H, improving strength development during and after the curing stage [44,70]. When adding large amounts of FA to a blended binder, mineral accelerators (e.g., CṠA and quicklime) are essential to improve the initial strength performance of the hardened backfill. CṠA hydrates rapidly to form large AFt phases to fill voids. Quicklime hydrates to increase the concentration of hydroxide ions that promote the activity of FA.

According to the manufacturer of raw materials, the CO_2_ emissions of SC, CṠA, FA, and quicklime products are 0.72, 0.69, 0.07, 0.18 kg/kg, and the costs are 71, 136.8, 23.4, and 32.1 $US/ton, respectively. The CO_2_ emission and cost of binders can be normalized per compressive strength with a 28-day curing time (Equations (3) and (4)), and the results are shown in Figure 15. It can be concluded that the SCC10 and SCC15 binders have lower CO_2_ emissions and costs than pure SC sample after 28 days of hydration. These experimental findings help to promote the practical application of blended binders in filling technology and optimize solid waste disposal.
(3)$$B_{{\mathrm{CO}}_{2}\;\mathrm{emission}}\;=\;\frac{{CO}_{2}\;emission}{UCS\;value\;}$$
(4)$$B_{cost}\;=\;\frac{cost}{UCS\;value\;}$$

## 4. Conclusions

In this study, a large amount of FA (40%) was used to lower the cost of the binder. Calcium sulfoaluminate cement and quicklime were used to accelerate the early hydration of blended binders. The flowability, mechanical performance, and pore structure of cemented paste backfill were examined. In addition, the heat of hydration and hydration evolution of pure blended binder pastes were investigated. Based on the experimental results, the conclusion can be drawn as:

(1) Adding FA significantly improves the flowability of the fresh CPB due to the lubrication effect of spherical shape. However, this increase was gradually suppressed by the increase of CṠA and quicklime. Due to the rapid reaction of CṠA, the SD decreased the fastest with increasing curing time in the SCA15 sample. As the amount of CṠA and quicklime increased, CPB samples with blended binders showed improved mechanical performance associated with the formation of ettringite and C–S–H/C–A–S–H. Increasing the dosage of quicklime from 10% to 15% slightly reduced the UCS value of the hardened backfill.

(2) The volume of pores with diameters less than 500 nm and 500–1000 nm did not show a clear trend with increasing amounts of CṠA and quicklime. In contrast, those larger than 1000 nm decreased dramatically. Linear fitting reveals that total pore volume was closely related to CPB strength. Due to the high content of cement clinker, the cumulative heat flow for the SC100 sample was high, and the addition of CṠA and quicklime increased the cumulative heat flow in the first 6 h.

(3) Addition of CṠA and quicklime promoted the formation of ettringite and portlandite. From day 7 to day 28 of hydration, FA was involved in hydration, thus reducing the amount of portlandite produced in blended binders. Furthermore, due to hydration of quicklime, FA was activated by the alkaline environment. Then, Al and Si released from FA were hydrated to form C–S–H/C–A–S–H with high Al/Si ratios.

(4) Blended binder SCC15 showed the least normalized CO_2_ emission and lower cost. Therefore, it had the potential for application in filling technology.

## Figures and Tables

**Figure 1 materials-13-04018-f001:**
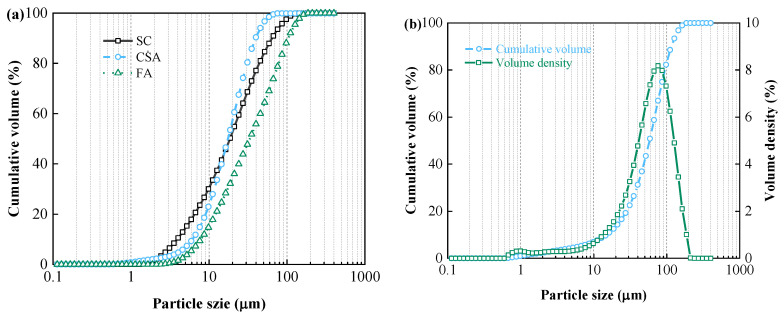
Particle size distribution of (**a**) Slag cement (SC), CṠA, and fly ash (FA); (**b**) silica tailings (ST).

**Figure 2 materials-13-04018-f002:**
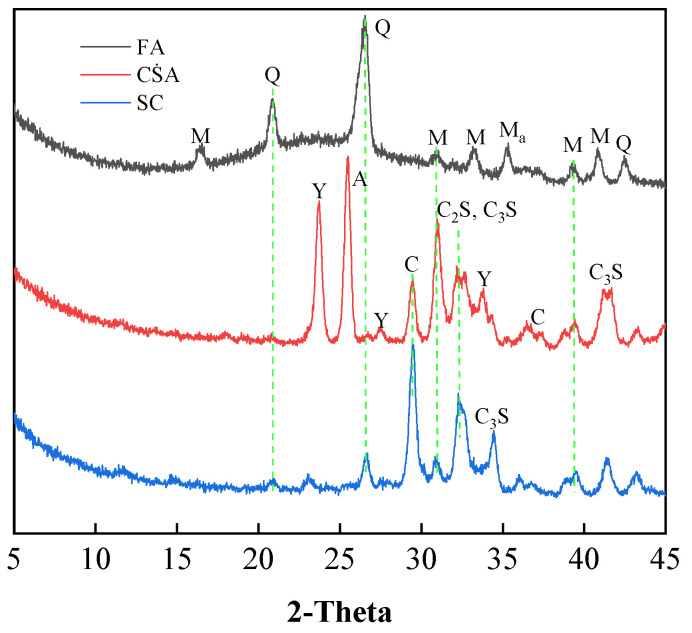
XRD patterns of FA, CṠA, and SC (Q—quartz, A—Anhydrite, C—calcite, Y—ye’elimite, M—mullite, M_a_—magnetite).

**Figure 3 materials-13-04018-f003:**
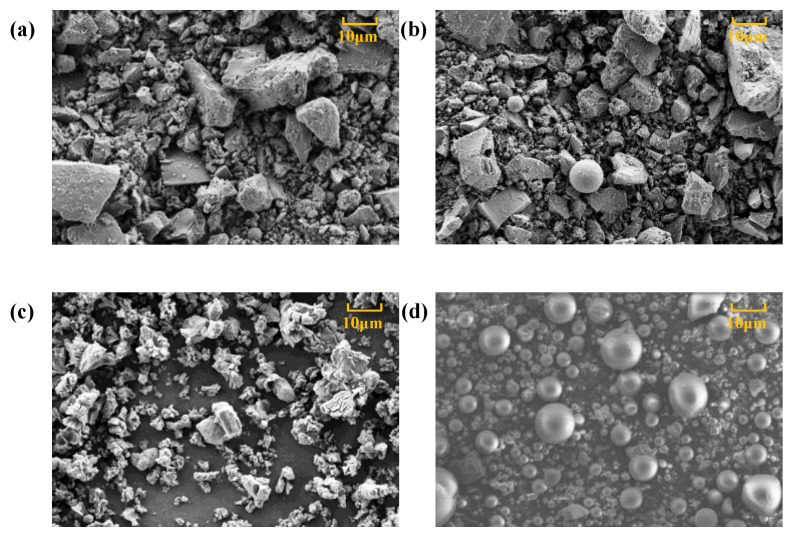
SEM images of (**a**) SC, (**b**) CṠA, (**c**) quicklime, and (**d**) FA (Mag = 3000×).

**Figure 4 materials-13-04018-f004:**
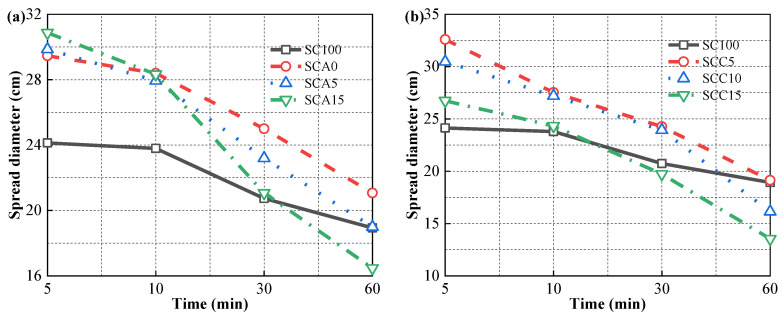
Spread diameter of fresh cemented paste backfill (CPB) with different doses of (**a**) CṠA and (**b**) quicklime.

**Figure 5 materials-13-04018-f005:**
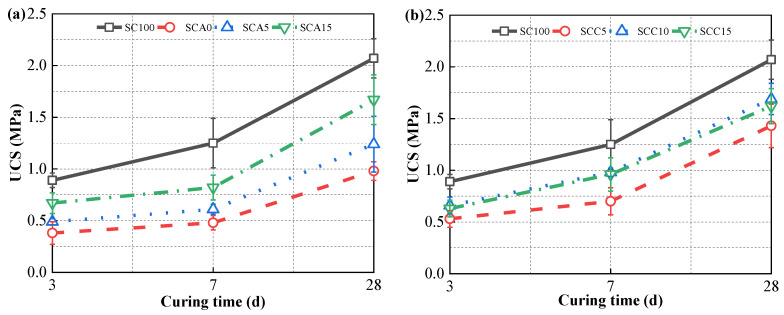
Unconfined compressive strength (UCS) variation of hardened CPB with different doses of (**a**) CṠA and (**b**) quicklime.

**Figure 6 materials-13-04018-f006:**
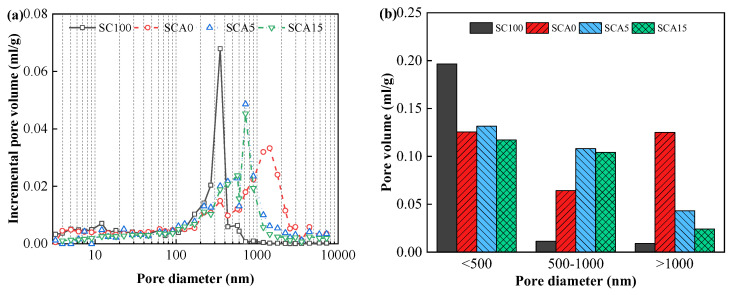
Pore size distribution results of hardened CPB with different doses of CṠA at the hydration time of 28 days: (**a**) the incremental pore volume, and (**b**) the volume of different pore types.

**Figure 7 materials-13-04018-f007:**
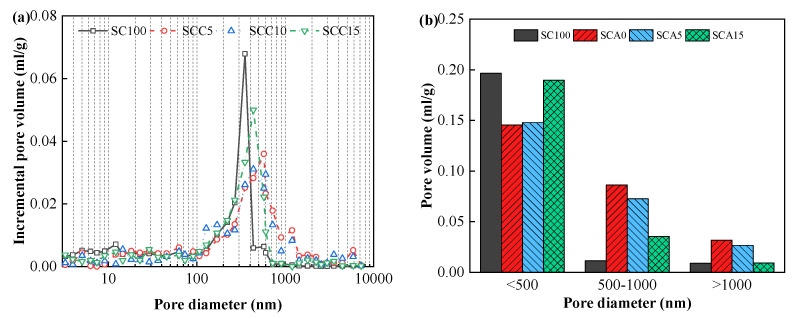
Pore size distribution results of hardened CPB with different doses of quicklime at the hydration time of 28 days: (**a**) the incremental pore volume, and (**b**) the volume of different pore types.

**Figure 8 materials-13-04018-f008:**
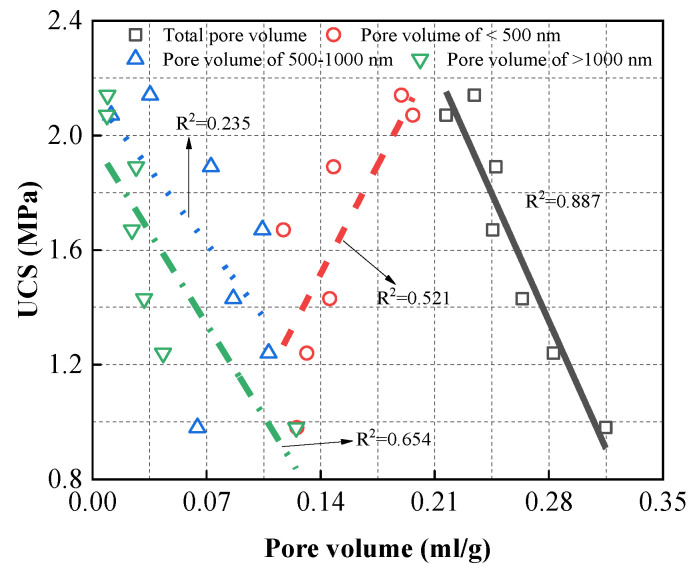
Relationship between UCS and pore volume.

**Figure 9 materials-13-04018-f009:**
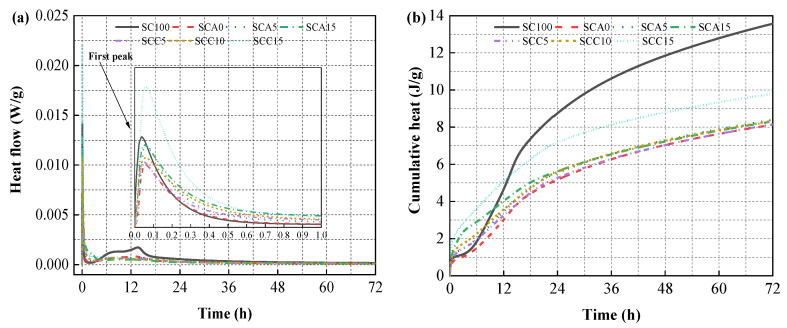
The heat flow of binder pastes: (**a**) first exothermic peak, and (**b**) cumulative heat.

**Figure 10 materials-13-04018-f010:**
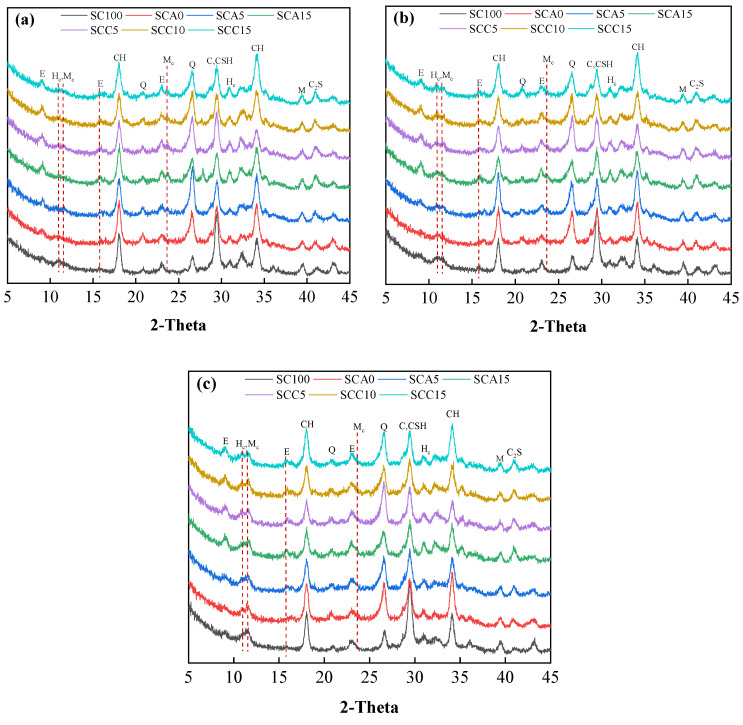
XRD patterns of binder pastes at hydration times of (**a**) 3 days, (**b**) 7 days, and (**c**) 28 days. (E = ettringite, H_c_ = hemicarbonaluminate, M_c_ = monocarbonaluminate, CH = portlandite, Q = quartz, C = Calcite, M = mullite, C_2_S = dicalcium silicate).

**Figure 11 materials-13-04018-f011:**
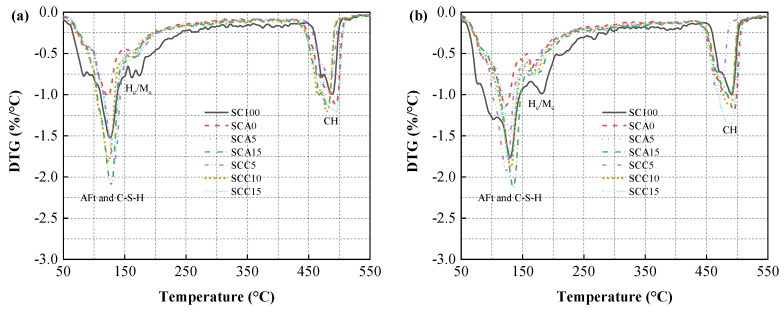
Differential thermogravimetry (DTG) curves of blended binder pastes at hydration times of (**a**) 3 days, (**b**) 7 days, and (**c**) 28 days.

**Figure 12 materials-13-04018-f012:**
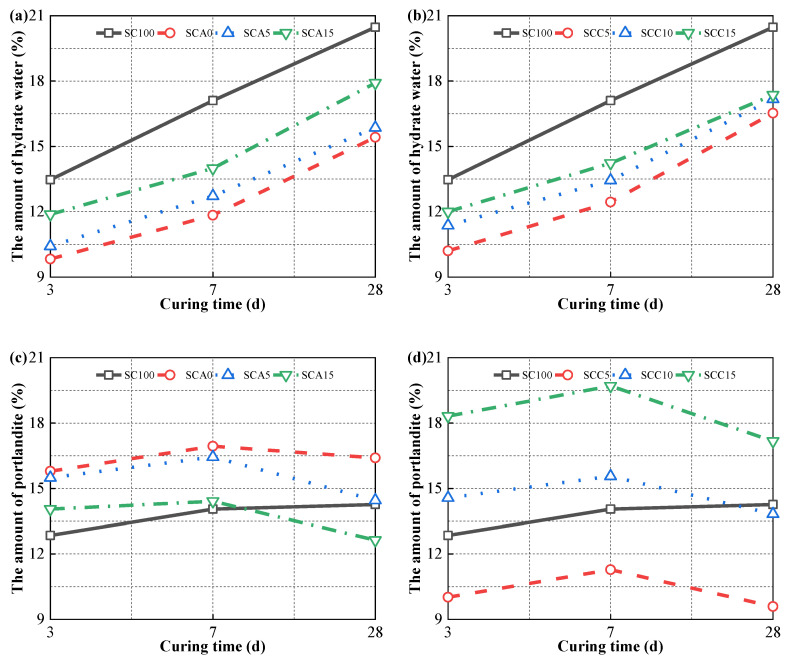
The content of hydrate water of binder pastes with different doses of (**a**) CṠA and (**b**) quicklime, and the content of portlandite of binder pastes with different doses of (**c**) CṠA and (**d**) quicklime.

**Figure 13 materials-13-04018-f013:**
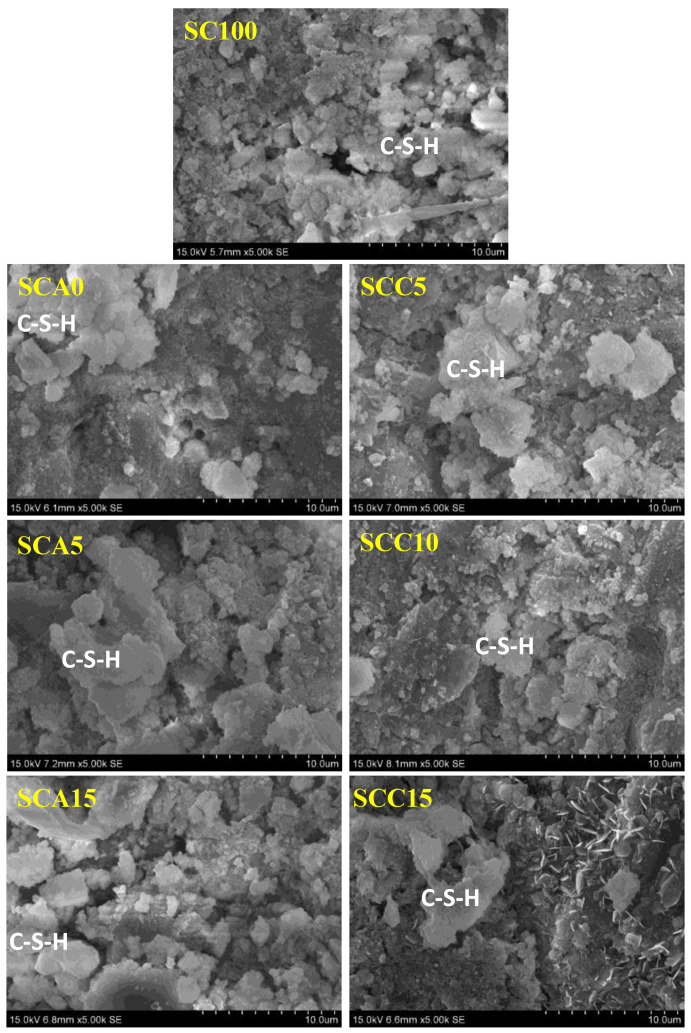
SEM observation of binder pastes after 28 days of hydration (Mag = 5000×).

**Figure 14 materials-13-04018-f014:**
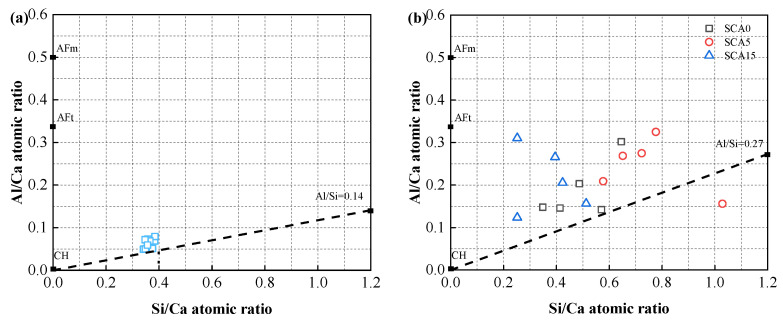
Plot of Al/Ca atomic ratio vs. Si/Ca atomic ratio of (**a**) SC100; (**b**) SCA0—SCA15 and (**c**) SCC5—SCC15 samples.

**Figure 15 materials-13-04018-f015:**
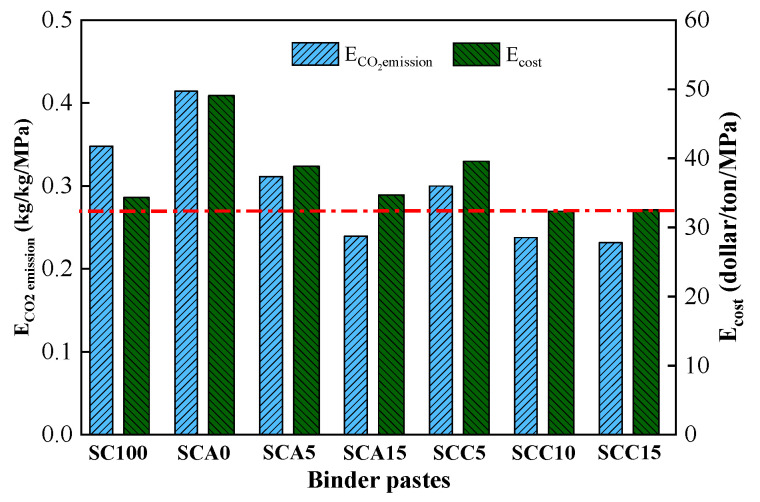
Normalized CO_2_ emission and cost of binder pastes.

**Table 1 materials-13-04018-t001:** Chemical composition of FA, CṠA, SC, and ST (%).

Element	SC	CṠA	FA	ST
CaO	55.24	53.91	4.08	0.03
SiO_2_	26.15	13.60	55.51	96.46
Al_2_O_3_	8.83	15.31	30.82	1.87
MgO	2.93	3.01	0.63	–
Fe_2_O_3_	3.02	2.03	4.02	0.07
Na_2_O	0.02	0.34	0.04	0.01
K_2_O	1.27	0.81	2.49	1.30
SO_2_	1.77	10.35	0.79	0.03
SSA ^a^ (m^2^/kg)	528.42	496.93	297.91	227.27

^a^ specific surface area.

**Table 2 materials-13-04018-t002:** Mixing ratios of blended binders (%).

Blended Binder	SC	CṠA	FA	Quicklime
SC100	100	0	0	0
SCA0	50	0	40	10
SCA5	45	5	40	10
SCA15	35	15	40	10
SCC5	45	10	40	5
SCC10	40	10	40	10
SCC15	35	10	40	15

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
