# Peer review of "Quicklime and Calcium Sulfoaluminate Cement Used as Mineral Accelerators to Improve the Properties of Cemented Paste Backfill with a High Volume of Fly Ash"

_materials, 2020, doi:10.3390/ma13184018_

Round 1
Reviewer 1 Report
The paper entitled "Quicklime and calcium sulfoaluminate cement used as mineral accelerators to improve the properties of cemented paste backfill with a high volume of fly ash" presents an interesting investigation in which a quicklime and calcium sulfoaluminate zre used to improve the properties of cemented paste.
The introduction is complete and the number of references is addecuate.
The methodology for its part is well structured and described.
In some of the titles capital letters have been used in all the words. It is recommended that only capital letters be used for proper names.
In the analysis of the microstructure, it is recommended to contrast the conclusions obtained from the analysis with some type of porosity analysis (gaseous or mercury intrusion) or CAT if possible.
For its part, the analysis of the results is adequate although it could go a little deeper in the aspects that are really relevant and are the strengths of the paper.
In my opinion the paper has a high scientific-technical quality and could be accepted after minor changes.
Reviewer 2 Report
The paper is generally well written, however, it contains some points that need to be discussed and improved. The reviewer would like to attach several comments as follows:
- Materials and methods
- Line 129 and 130: Please describe clearly the curing condition, for example, did the author cover the top surface of specimens?
- In the unconfined compression test: please indicate which standard the author used.
- Line 144-145: “At least two samples were measured for each recipe”, please state clearly how many specimens were used for each case (which case having 2 samples and which one having 3 samples).
- Pore structure: Please describe the treatment of the sample before conducting the test. For example, how the author dried samples, etc.
- Hydration evolution: Line 159: “vacuum drying oven to a constant weight”, which temperature the author used to dry the specimens?
- Line 165: why the author used 15°C/min, also please cite the reference.
- Equation (1) and (2), why the author compared to the mass of the residual at 550 oC, why the author did not compare with the mass of the residual at 1000 oC?
- Results
- Pore structures: Line 253-254, why the author can classify the pore size in those three groups? Please also consider some references (publications) as below:
- Analysis of strength development in cement-treated soils under different curing conditions through microstructural and chemical investigations
- Enhanced thermodynamic analysis coupled with temperature-dependent microstructures of cement hydrates
- Analysis of strength development in cement-stabilized silty clay from microstructural considerations
- TG analysis: line 338: “the decomposition temperatures of hydration products and portlandite are 50–550 °C and 400–550°C”, how is the decomposition after 550 oC?
- In the TG analysis, did the author consider the loss of raw material (cement, silica tailing …), and even the loss (decomposition) of unhydrated cement, or the decomposition of carbonate from raw cement?
Author Response
Comments:
The paper is generally well written, however, it contains some points that need to be discussed and improved. The reviewer would like to attach several comments as follows:
Q1. Line 129 and 130: Please describe clearly the curing condition, for example, did the author cover the top surface of specimens?
Response: Thanks for your kind suggestion. We have revised the statement of curing condition in the revised manuscript as below.
“It should be noted that both of the upper and lower ends of the moulds are sealed. All samples were cured at room temperature (20 ± 2 °C) with a relative humidity of 95 ± 2%.”
Q2. In the unconfined compression test: please indicate which standard the author used.
Response: Thanks for your constructive suggestion. We have added the standard used in unconfined compressive strength test in the revised manuscript as below.
“After reaching the pre-determined hydration time (3, 7, and 28 days), CPB samples were subjected to the UCS test based on ASTM C39 standard test procedure [37].”
The reference used in the above paragraph is given as below:
- Standard, A., Standard test method for compressive strength of cylindrical concrete specimens. ASTM C39 2010.
Q3. Line 144-145: “At least two samples were measured for each recipe”, please state clearly how many specimens were used for each case (which case having 2 samples and which one having 3 samples).
Response: Thanks for your constructive suggestions. We have corrected the sentence in the revised manuscript as below.
“If the measurements of the two samples differ by more than 15%, a third sample needs to be measured to verify its accuracy. Hence, at least two samples were measured for each recipe, and only the mean value was considered the UCS for CPB samples.”
Q4. Pore structure: Please describe the treatment of the sample before conducting the test. For example, how the author dried samples, etc.
Response: Thanks for your constructive suggestion. We have added the detailed treatment of samples in the revised manuscript as below.
“Samples were immersed in isopropanol to stop the hydration for 12 h, and then dried in vacuum drying oven at 45°C for 24 h.”
Q5. Hydration evolution: Line 159: “vacuum drying oven to a constant weight”, which temperature the author used to dry the specimens?
Response: Thanks for your suggestion. We have modified the statement in the revised manuscript as below.
“Samples were pulverized after drying in a vacuum drying oven to a constant weight at 45 °C.”
Q6. Line 165: why the author used 15°C/min, also please cite the reference.
Response: We agree with the reviewer that some important references are missing and now they have been added to support the statement in the revised manuscript.
The references used in the paragraph are listed as below:
- Zhao, Y.; Qiu, J.; Xing, J.; Sun, X., Chemical activation of binary slag cement with low carbon footprint. Journal of Cleaner Production 2020, 267, 121455.
- Zhao, Y.; Qiu, J.; Zhang, S.; Guo, Z.; Ma, Z.; Sun, X.; Xing, J., Effect of sodium sulfate on the hydration and mechanical properties of lime-slag based eco-friendly binders. Construction and Building Materials 2020, 250, 118603.
Q7. Equation (1) and (2), why the author compared to the mass of the residual at 550 °C, why the author did not compare with the mass of the residual at 1000 °C?
Response: Thanks for pointing out this issue. We normalize the content of hydrate water and portlandite in binder pastes according to the residual mass at the critical maximum temperature (550°C) of portlandite decomposition.
Q8. Results
Pore structures: Line 253-254, why the author can classify the pore size in those three groups? Please also consider some references (publications) as below:
Analysis of strength development in cement-treated soils under different curing conditions through microstructural and chemical investigations
Enhanced thermodynamic analysis coupled with temperature-dependent microstructures of cement hydrates
Analysis of strength development in cement-stabilized silty clay from microstructural considerations
Response: Thanks for reviewer’s constructive suggestions. We have improved the statement of classification of pore size in the revised manuscript as below.
“Based on the references [50-53], pores with diameter smaller than 500 nm are micropores and mespories, and pores with diameter larger than 1000 nm are large capillary pores. Hence, pores are classified into three types according to the pore size in this study: (1) type I: <500 nm; (2) type II: 500-1000 nm; (3) type III: >1000 nm.”
The references used in the paragraph are listed as below:
- Everett, D., " IUPAC Manual of Symbols and Terminology", appendix 2, Part 1, Colloid and Surface Chemistry. Pure Appl. Chem. 1972, 31, 578-621.
- Ho, L. S.; Nakarai, K.; Duc, M.; Le Kouby, A.; Maachi, A.; Sasaki, T., Analysis of strength development in cement-treated soils under different curing conditions through microstructural and chemical investigations. Construction and Building Materials 2018, 166, 634-646.
- Nakarai, K.; Ishida, T.; Kishi, T.; Maekawa, K., Enhanced thermodynamic analysis coupled with temperature-dependent microstructures of cement hydrates. Cement and concrete research 2007, 37, (2), 139-150.
- Horpibulsuk, S.; Rachan, R.; Chinkulkijniwat, A.; Raksachon, Y.; Suddeepong, A., Analysis of strength development in cement-stabilized silty clay from microstructural considerations. Construction and building materials 2010, 24, (10), 2011-2021.
Q9. TG analysis: line 338: “the decomposition temperatures of hydration products and portlandite are 50–550 °C and 400–550°C”, how is the decomposition after 550° C?
Response: Thanks for your kind suggestion. We have corrected the statement in the revised manuscript as below.
“According to the literatures [40, 61, 62], the decomposition temperatures of hydration products (AFt, AFm, and C-(A)-S-H) and portlandite are 50–550 °C and 400–550°C, respectively, while the decomposition of carbonate mainly appears between 550–1000°C.”
The references used in the paragraph are listed as below:
- De Weerdt, K.; Haha, M. B.; Le Saout, G.; Kjellsen, K. O.; Justnes, H.; Lothenbach, B., Hydration mechanisms of ternary Portland cements containing limestone powder and fly ash. Cement and Concrete Research 2011, 41, (3), 279-291.
- Bullard, J. W.; Jennings, H. M.; Livingston, R. A.; Nonat, A.; Scherer, G. W.; Schweitzer, J. S.; Scrivener, K. L.; Thomas, J. J., Mechanisms of cement hydration. Cement and Concrete Research 2011, 41, (12), 1208-1223.
- Briendl, L. G.; Mittermayr, F.; Baldermann, A.; Steindl, F. R.; Sakoparnig, M.; Letofsky-Papst, I.; Galan, I., Early hydration of cementitious systems accelerated by aluminium sulphate: Effect of fine limestone. Cement and Concrete Research 2020, 134, 106069.
Q10. In the TG analysis, did the author consider the loss of raw material (cement, silica tailing …), and even the loss (decomposition) of unhydrated cement, or the decomposition of carbonate from raw cement?
Response: Thanks for pointing out this issue. The content of hydrate water and portlandite are calculated by the weight loss of pure binder pastes between 50 and 550 °C, during which the decomposition of raw unhydrated cement almost has no effect on the TG results.

Reviewer 3 Report
This manuscript presents the results of novel blended binders of Slag Cement (SC), Fly Ash (FA), Quicklime, Calcium Sulfoaluminate (CSA). Where CSA and quicklime were used to accelerate the hydration of blended binders and FA was used to reduce the cost of the binder and CO2 reduce emissions. The topic is interesting. However, there are some points that would be good to correct:
Line 95: The chemical composition of CSA in line 95 (53.19%), does not correspond to that presented in the chemical composition of CSA shown in table 1 (53.91%).
Line 116: Scale bar should be added to micrographs, to have a comparison of measurements between the particles presented in each micrographs.
Line 148: in "15 x 15 x 15 mm3", it could be: 15 x 15 x 15 mm, or 3375 mm3.
Line 209-210: What was the reference (value) to comment that there was an increase in values of 0.21 and 0.39 MPa, with SCA5 and SCA15 samples, respectively. You could better explain these two lines (209 and 210).
Line 325: In the figure caption of figure 10, it is necessary to add C2S and its meaning.
Line 365: In the figure caption of figure 12, detail each micrograph (a, b, c and d).
Line 409: check units, "kg / kg"
Round 2
Reviewer 2 Report
Thank you very much for addressed all comments of reviewers.
The paper has been improved significantly.
Now it can be accepted for publication.
Thank you.